# Offline-to-Online Reinforcement Learning via Balanced Replay and Pessimistic Q-Ensemble

**Seunghyun Lee**[*][†]     **Younggyo Seo**[*][†]     **Kimin Lee**[‡]     **Pieter Abbeel**[‡]     **Jinwoo Shin**[†]
[†] Korea Advanced Institute of Science and Technology
[‡]University of California, Berkeley

**Abstract:** Recent advance in deep offline reinforcement learning (RL) has made it possible to train strong robotic agents from offline datasets. However, depending on the quality of the trained agents and the application being considered, it is often desirable to fine-tune such agents via further online interactions. In this paper, we observe that state-action distribution shift may lead to severe bootstrap error during fine-tuning, which destroys the good initial policy obtained via offline RL. To address this issue, we first propose a balanced replay scheme that prioritizes samples encountered online while also encouraging the use of near-on-policy samples from the offline dataset. Furthermore, we leverage multiple Q-functions trained pessimistically offline, thereby preventing overoptimism concerning unfamiliar actions at novel states during the initial training phase. We show that the proposed method improves sample-efficiency and final performance of the fine-tuned robotic agents on various locomotion and manipulation tasks. Our code is available at: https://github.com/shlee94/Off2OnRL.

**Keywords:** Deep Reinforcement Learning, Offline RL, Fine-tuning

## 1 Introduction

Deep offline reinforcement learning (RL) [1] has the potential to train strong robotic agents without any further environment interaction by leveraging deep neural networks and huge offline datasets. Accordingly, the research community has demonstrated that offline RL can train both simulated [2, 3, 4, 5, 6, 7, 8, 9] and real [5, 10] robots that are often more performant than the behavior policy that generated the offline dataset. However, thusly trained offline RL agents may be suboptimal, for (a) the dataset they were trained on may be suboptimal; and (b) environment in which they are deployed may be different from the environment in which the dataset was generated. This necessitates an online fine-tuning procedure, where the robot improves by gathering additional samples.

Off-policy RL algorithms are well-suited for offline-to-online RL, since they can leverage both offline and online samples. Fine-tuning an offline RL agent using a conventional off-policy RL algorithm, however, is difficult due to distribution shift, i.e., the robot may encounter unfamiliar state-action regime that is not covered by the offline dataset. The Q-function cannot provide accurate value estimates for such out-of-distribution (OOD) online samples, and updates with such samples lead to severe bootstrap error. This leads to policy updates in an arbitrary direction, destroying the good initial policy obtained by offline RL.

To address state-action distribution shift, we first introduce a balanced replay scheme that enables us to provide the robotic agent with near-on-policy samples from the offline dataset, in addition to samples gathered online. Specifically, we train a network that measures the *online-ness* of available samples, then prioritize samples according to this measure. This adjusts the sampling distribution for Q-learning to be closer to online samples, which enables timely value propagation and more accurate policy evaluation in the novel state-action regime.

However, we find that the above sampling scheme is not enough, for the Q-function may be overoptimistic about unseen actions at novel online states. This misleads the robot to prefer potentially

---

[*]Equal Contribution. Correspondence to {seunghyun.lee, younggyo.seo}@kaist.ac.kr

5th Conference on Robot Learning (CoRL 2021), London, UK.

bad actions, and in turn, more severe distribution shift and bootstrap error. We therefore propose a pessimistic Q-ensemble scheme. In particular, we first observe that a specific class of offline RL algorithms that train pessimistic Q-functions [8, 9] make an excellent starting point for offline-to-online RL. When trained as such, the Q-function implicitly constrains the policy to stay near the behavior policy during the initial fine-tuning phase. Building on this observation, we leverage multiple pessimistic Q-functions, which guides the robotic agent with a more high-resolution pessimism and stabilizes fine-tuning.

In our experiments, we demonstrate the strength of our method based on (1) MuJoCo [11] locomotion tasks from the D4RL [12] benchmark suite, and (2) vision-based robotic manipulation tasks from Singh et al. [10]. We show that our method achieves stable training during fine-tuning, while outperforming all baseline methods considered, both in terms of final performance and sample-efficiency. We provide a thorough analysis of each component of our method.

## 2  Background

**Reinforcement learning.** We consider the standard RL framework, where an agent interacts with the environment so as to maximize the expected total return. More formally, at each timestep $t$, the agent observes a state $s_t$, and performs an action $a_t$ according to its policy $\pi$. The environment rewards the agent with $r_t$, then transitions to the next state $s_{t+1}$. The agent's objective is to maximize the expected return $\mathbb{E}_\pi[\sum_{t=0}^\infty \gamma^t r_t]$, where $\gamma \in [0, 1)$ is the discount factor. The unnormalized stationary state-action distribution under $\pi$ is defined as $d^\pi(s, a) := \sum_{t=0}^\infty \gamma^t d_t^\pi(s, a)$, where $d_t^\pi(s, a)$ denotes the state-action distribution at timestep $t$ of the Markov chain defined by the fixed policy $\pi$.

**Soft actor-critic.** We mainly consider off-policy RL algorithms, a class of algorithms that can, in principle, train an agent with samples generated by any behavior policy. In particular, soft actor-critic [SAC; 13] is an off-policy actor-critic algorithm that learns a soft Q-function $Q_\theta(s, a)$ parameterized by $\theta$ and a stochastic policy $\pi_\phi$ modeled as a Gaussian, parameterized by $\phi$. SAC alternates between critic and actor updates by minimizing the following objectives, respectively:

$$\mathcal{L}_{\texttt{critic}}^{\texttt{SAC}}(\theta) = \mathop{\mathbb{E}}_{(s,a,s')\sim\mathcal{B}}\left[\left(Q_\theta(s,a) - r(s,a) - \gamma\mathop{\mathbb{E}}_{a'\sim\pi_\phi}\left[Q_{\bar\theta}(s',a') - \alpha\log\pi_\phi(a'|s')\right]\right)^2\right], \quad (1)$$

$$\mathcal{L}_{\texttt{actor}}^{\texttt{SAC}}(\phi) = \mathop{\mathbb{E}}_{s\sim\mathcal{B},a\sim\pi_\phi}\left[\alpha\log\pi_\phi(a|s) - Q_\theta(s,a)\right], \quad (2)$$

where $\mathcal{B}$ is the replay buffer, $\bar\theta$ the delayed parameters, and $\alpha$ the temperature parameter.

**Conservative Q-learning.** Offline RL algorithms are off-policy RL algorithms that utilize static datasets for training an agent. In particular, conservative Q-learning [CQL; 9] pessimistically evaluates the current policy, and learns a lower bound (in expectation) of the ground-truth Q-function. To be specific, policy evaluation step of CQL minimizes the following:

$$\mathcal{L}_{\texttt{critic}}^{\texttt{CQL}}(\theta) = \frac{1}{2}\mathop{\mathbb{E}}_{(s,a,s')\sim\mathcal{B}}\left[(Q_\theta - \mathcal{B}^{\pi_\phi}Q_{\bar\theta})^2\right] + \alpha_0\mathop{\mathbb{E}}_{s\sim\mathcal{B}}\left[\log\sum_a\exp Q(s,a) - \mathop{\mathbb{E}}_{a\sim\hat\pi_\beta}[Q(s,a)]\right], \quad (3)$$

where $\hat\pi_\beta(a_0|s_0) := \frac{\sum_{s,a\in\mathcal{B}}\mathbb{1}[s=s_0,a=a_0]}{\sum_{s\in\mathcal{B}}\mathbb{1}[s=s_0]}$ is the empirical behavior policy, $\alpha_0$ the trade-off factor, and $\mathcal{B}^\pi$ the bellman operator. The first term is the usual Bellman backup, and the second term is the regularization term that decreases the Q-values for unseen actions, while increasing the Q-values for seen actions. We argue that thusly trained pessimistic Q-function is beneficial for fine-tuning as well (see Figure 1c). Policy improvement step is the same as SAC defined in (2).

## 3  Fine-tuning Offline RL Agent

In this section, we investigate the distribution shift problem in offline-to-online RL. We first explain why an agent being fine-tuned can be susceptible to distribution shift, and why distribution shift is problematic. Then, we demonstrate two important design choices that decide the effect of distribution shift on fine-tuning: *sample selection* and *choice of offline Q-function*.

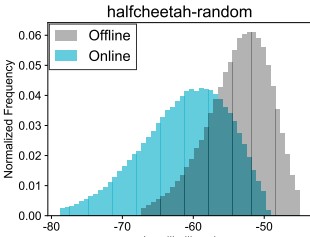
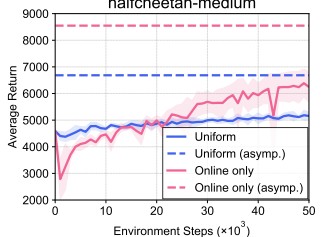
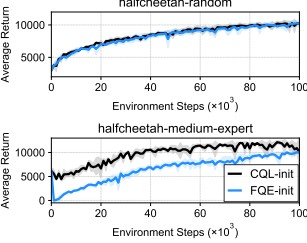

| (a) State-action distribution shift | (b) Sample selection | (c) Choice of offline Q-function |

Figure 1: (a) Log-likelihood estimates of (i) offline samples and (ii) online samples gathered by the offline RL agent, based on a VAE model trained on the offline dataset. (b) Fine-tuning performance on `halfcheetah-medium` task when using online samples exclusively (**Online only**), or when using both offline and online data drawn uniformly at random (**Uniform**). (c) Fine-tuning performance on `halfcheetah-random` and `halfcheetah-medium-expert` tasks, when using a pessimistic (denoted **CQL-init**) and a non-pessimistic (denoted **FQE-init**) Q-function, respectively.

## 3.1 Distribution Shift in Offline-to-Online RL

In offline-to-online RL, there exists a distribution shift between $d^{\text{on}}(s, a)$ and $d^{\text{off}}(s, a)$, where the former denotes the state-action distribution of online samples in the online buffer $\mathcal{B}^{\text{on}}$, and the latter that of offline samples in the offline buffer $\mathcal{B}^{\text{off}}$. Figure 1a visualizes such distribution shift. Specifically, we trained a variational autoencoder [14] to reconstruct state-action pairs in the `halfcheetah-random` dataset that contains uniform random policy rollouts in the halfcheetah environment. Then, we compared the log-likelihood of (a) offline samples and (b) online samples collected by a CQL agent trained on the same dataset. There is a clear difference between offline and online state-action distributions.

Such distribution shift is problematic, for the agent will enter the unseen state-action regime, where Q-values (hence value estimates used for bootstrapping) can be very inaccurate. Updates in such unseen regime results in erroneous policy evaluation and arbitrary policy updates, which destroys the good initial policy obtained via offline RL. Distribution shift can be especially severe in offline-to-online RL, for the offline RL agent is often much more performant than the behavior policy (e.g., CQL can train a medium-level agent capable of running, using transitions generated by a random policy only). Also, when the offline dataset is narrowly distributed, e.g., when it is generated by a single policy, the agent is more prone to distribution shift, for the agent easily deviates from the narrow, seen distribution.

## 3.2 Sample Selection

In light of the above discussion, we study how sample selection affects fine-tuning. We find that online samples, which are essential for fine-tuning, are also potentially dangerous OOD samples due to distribution shift. Meanwhile, offline samples are in-distribution and safe, but leads to slow fine-tuning. As a concept experiment, we trained an agent offline via CQL (3, 2) on the `halfcheetah-medium` dataset containing medium-level transitions, then fine-tuned the agent via SAC (1, 2). We see that using online samples exclusively for updates (denoted **Online only** in Figure 1b) leads to unstable fine-tuning, where the average return drops from about 4500 to below 3000. This demonstrates the harmful effect of distribution shift, where novel, OOD samples collected online cause severe bootstrap error.

On the other hand, when using a single replay buffer for both offline and online samples then sampling uniformly at random (denoted **Uniform** in Figure 1b), the agent does not use enough online samples for updates, especially when the offline dataset is large. As a result, value propagation is slow, and as seen in Figure 1b, this scheme achieves initial stability at the cost of asymptotic performance. This motivates a balanced replay scheme that modulates the trade-off between using online samples (useful, but potentially dangerous), and offline samples (stable, but slow fine-tuning).

### 3.3 Choice of Offline Q-function

Another important design choice in offline-to-online RL is the offline training of Q-function. In particular, we show that a pessimistically trained Q-function mitigates the effect of distribution shift, by staying conservative about OOD actions in the initial training phase. As a concept experiment, we compared the fine-tuning performance when using a pessimistically trained Q-function and when using a Q-function trained without any pessimistic regularization. Specifically, for a given offline dataset, we first trained a policy $\pi_\phi$ and its pessimistic Q-function $Q_{\theta_{\text{CQL}}}$ via CQL (3). Then we trained a non-pessimistic Q-function $Q_{\theta_{\text{FQE}}}$ of the pre-trained offline policy $\pi_\phi$ via Fitted Q Evaluation [FQE; 15], an off-policy policy evaluation method that trains a given policy's Q-function. Finally, we fine-tuned $\{\pi_\phi, Q_{\theta_{\text{CQL}}}\}$ and $\{\pi_\phi, Q_{\theta_{\text{FQE}}}\}$ via SAC (1, 2). See Section C for more details.

As shown in Figure 1c, both pessimistic and non-pessimistic Q-functions show similar fine-tuning performance on the `random` dataset, which contains random policy rollouts with good action space coverage. However, when fine-tuning an offline RL agent trained on the `medium-expert` dataset, which contains transitions obtained by a mixture of more selective and performant policies, non-pessimistic Q-function loses the good initial policy, reaching zero average return at one point.

The reason is that $Q_{\text{FQE}}$ can be overly optimistic about OOD actions at novel states when bootstrapping from them. In turn, the policy may prefer potentially bad actions, straying further away from the safe, seen trajectory. On the other hand, $Q_{\text{CQL}}$ remains pessimistic in the states encountered online initially, for (1) these states are incrementally different from seen states, and (2) Q-function will thus have similar pessimistic estimates due to generalization. This points to a strategy where we first train a pessimistic Q-function offline, then let it gradually lose the pessimism as the agent gains access to a balanced mix of offline and online samples via balanced replay during fine-tuning. Furthermore, since a single agent's Q-function may not be pessimistic enough, we may train multiple agents offline in parallel, then deploy online the ensemble agent equipped with a higher-resolution pessimism of the Q-ensemble (see Section 4.2 for a more detailed explanation).

## 4 Method

We propose a simple yet effective framework that addresses the state-action distribution shift described in Section 3. Our method comprises of two parts: (a) a balanced experience replay scheme, and (b) a pessimistic Q-ensemble scheme.

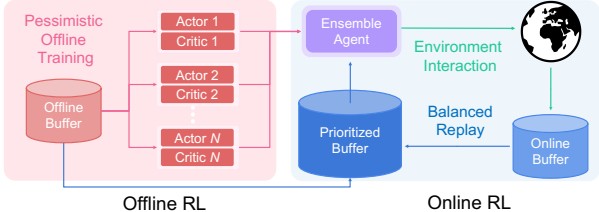

### 4.1 Balanced Experience Replay

We introduce a balanced replay scheme that enables us to safely utilize online samples by leveraging relevant, near-on-policy offline samples. By doing so, we can widen the sampling distribution for updates around the on-policy samples and enable timely value propaga-

Figure 2: Illustration of our framework. We first train an ensemble of $N$ CQL agents on the offline dataset. Then we fine-tune the ensemble agent using both offline and online transitions via balanced replay. In particular, we train a density ratio estimator that measures the online-ness of a given sample, then store all samples in the prioritized replay buffer with their respective density ratios as priority values. In turn, samples are drawn with probability proportional to their respective priority values.

tion. The challenge here is how to design a scheme that locates and retrieves such relevant, near-on-policy samples from the offline dataset, which can often be huge. To achieve this, we measure the *online-ness* of all available samples, and prioritize the samples according to this measure.

In particular, when updating the agent, we propose to sample a transition $(s, a, s') \in \mathcal{B}^{\text{off}} \cup \mathcal{B}^{\text{on}}$ with a probability proportional to the density ratio $w(s, a) := d^{\text{on}}(s, a)/d^{\text{off}}(s, a)$ of the given sample. This way, we can retrieve a relevant, near-on-policy sample $(s, a, s') \in \mathcal{B}^{\text{off}}$ by locating a transition with high density ratio $w(s, a)$. However, estimating the likelihoods $d^{\text{off}}(s, a)$ and $d^{\text{on}}(s, a)$ is difficult, since they can in principle be stationary distributions of complex policy mixtures[2]. To avoid this problem, we utilize a likelihood-free density ratio estimation method that estimates $w(s, a)$ by training a network $w_\psi(s, a)$ parametrized by $\psi$, solely based on samples from $\mathcal{B}^{\text{off}}$ and $\mathcal{B}^{\text{on}}$.

---

[2]We remark that $d^{\text{off}}(s, a)$ is the stationary distribution of the (arbitrary) behavior policy that generated $\mathcal{B}^{\text{off}}$, and $d^{\text{on}}(s, a)$ the stationary distribution of the policy that generated $\mathcal{B}^{\text{on}}$, which corresponds to the mixture of online policies observed over the course of fine-tuning.

**Training details.** Here we describe the training procedure for the density ratio estimator $w_\psi(s, a)$ in detail. Following the idea of Sinha et al. [16], we use the variational representation of f-divergences [17]. Let $P$ and $Q$ be probability measures defined on some measurable space $\mathcal{X}$, with $P$ absolutely continuous w.r.t. $Q$, and $f(y) := y \log \frac{2y}{y+1} + \log \frac{2}{y+1}$. Then the Jensen-Shannon (JS) divergence is defined as $D_{JS}(P||Q) = \int_\mathcal{X} f(dP(x)/dQ(x))dQ(x)$. We then estimate the density ratio $dP/dQ$ with a parametric model $w_\psi(x)$, by maximizing the lower bound of $D_{JS}(P||Q)$ [17]:

$$\mathcal{L}^{\mathrm{DR}}(\psi) = \mathbb{E}_{x \sim P}[f'(w_\psi(x))] - \mathbb{E}_{x \sim Q}[f^*(f'(w_\psi(x)))], \tag{4}$$

where $w_\psi(x) \geq 0$ is parametrized by a neural network whose outputs are forced to be non-negative via activation functions, and $f^*$ denotes convex conjugate. In particular, we obtain an estimate $w_\psi(s, a)$ of $d^{\mathrm{on}}/d^{\mathrm{off}}$ by considering probability distributions $P$ and $Q$ with densities $d^{\mathrm{on}}$ and $d^{\mathrm{off}}$, respectively. In practice, we sample from $\mathcal{B}^{\mathrm{on}}$ for the first term in (4), and from $\mathcal{B}^{\mathrm{off}}$ for the latter. For more stable density ratio estimates, we apply self-normalization [18] to the estimated density ratios over $\mathcal{B}^{\mathrm{off}}$, similar to Sinha et al. [16]. More details can be found in Section D.

## 4.2 Pessimistic Q-Ensemble

In order to mitigate distribution shift more effectively, we leverage multiple pessimistically trained Q-functions. We consider an ensemble of $N$ CQL agents pre-trained via update rules (2, 3), i.e., $\{Q_{\theta_i}, \pi_{\phi_i}\}_{i=1}^N$, where $\theta_i$ and $\phi_i$ denote the parameters of the $i$-th agent's Q-function and policy, respectively. Then we use the ensemble of actor-critic agents whose Q-function and policy are defined as follows:

$$Q_\theta := \frac{1}{N} \sum_{i=1}^N Q_{\theta_i}, \quad \pi_\phi(\cdot|s) = \mathcal{N}\left(\frac{1}{N} \sum_{i=1}^N \mu_{\phi_i}(s), \quad \frac{1}{N} \sum_{i=1}^N (\sigma_{\phi_i}^2(s) + \mu_{\phi_i}^2(s)) - \mu_\phi^2(s)\right), \tag{5}$$

where $\theta := \{\theta_i\}_{i=1}^N$ and $\phi := \{\phi_i\}_{i=1}^N$. Note that the policy is simply modeled as Gaussian with mean and variance of the Gaussian mixture policy $\frac{1}{N} \sum_{i=1}^N \pi_{\phi_i}$. In turn, $\theta$ and $\phi$ are updated via update rules (1) and (2), respectively, during fine-tuning.

By using a pessimistic Q-function, the agent remains pessimistic with regard to the unseen actions at states encountered online during initial fine-tuning. This is because during early fine-tuning, states resemble those present in the offline dataset, and Q-function generalizes to these states. As we show in our experiments, this protects the good initial policy from severe bootstrap error. And by leveraging multiple pessimistically trained Q-functions, we obtain a more high-resolution pessimism about the unseen data regime. That is, when an individual Q-function may erroneously have high values for unseen samples, Q-ensemble is more robust to these individual errors, and more reliably assigns lower values to unseen samples. Computational overhead of ensemble is discussed in Section B.

## 5 Related work

**Offline RL.** Offline RL algorithms aim to train RL agents exclusively with pre-collected datasets. To address the state-conditional action distribution shift, prior methods (a) explicitly constrain the policy to be closed to the behavior policy [2, 3, 4, 5, 19], or (b) train pessimistic value functions [7, 8, 9]. In particular, CQL [9] was used to learn various robotic manipulation tasks [10]. We also build on CQL, so as to leverage pessimism regarding data encountered online during fine-tuning.

**Online RL with offline datasets.** Several works have explored employing offline datasets for online RL to improve sample efficiency. Some assume access to demonstration data [20, 21, 22, 23, 24], which is limited in that they assume optimality of the dataset. To overcome this, Nair et al. [25] proposed Advantage Weighted Actor Critic (AWAC), which performs regularized policy updates so that the policy stays close to the observed data during both offline and online phases. We instead advocate adopting pessimistic *initialization*, such that we may prevent overoptimism and bootstrap error in the initial online phase, and lift such pessimism once unnecessary, as more online samples are gathered. Some recent works extract behavior primitives from offline data, then learn to compose them online [26, 27, 28]. It would be interesting to apply our method in these setups.

**Experience replay.** The idea of retrieving important samples for RL was introduced in Schaul et al. [29], where they prioritize samples with high temporal-difference error. The work closest to ours is Sinha et al. [16], which utilizes the density ratios between off-policy and near-on-policy state-action distributions as importance weights for policy evaluation. Our approach differs in that we utilize density ratios for retrieving relevant samples from the offline dataset.

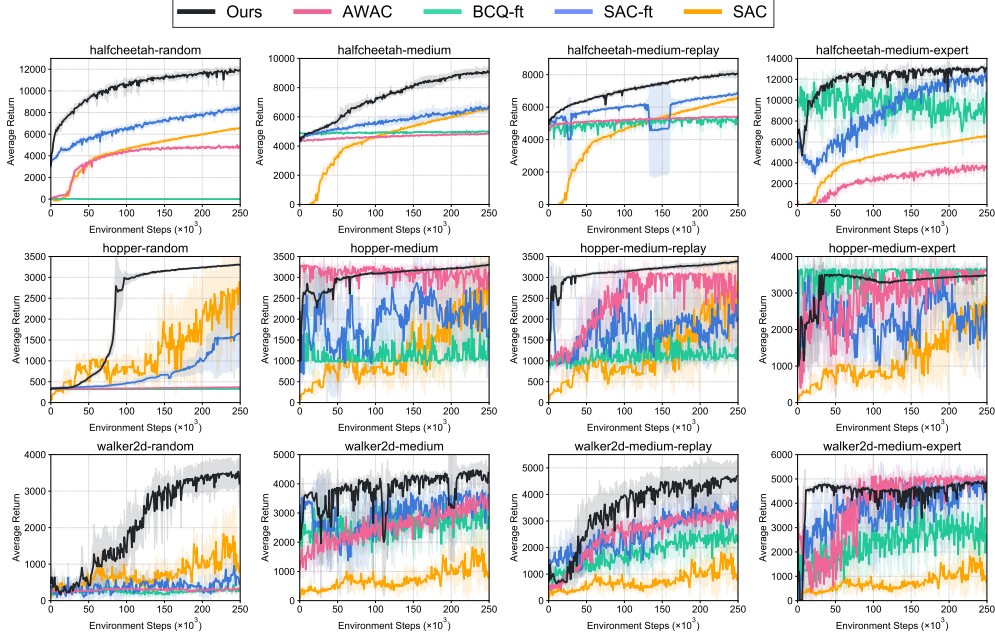

Figure 3: Performance on D4RL [12] MuJoCo locomotion tasks during online fine-tuning. The solid lines and shaded regions represent mean and standard deviation, respectively, across four runs.

**Ensemble methods.** In the context of model-free RL, ensemble methods have been studied for addressing Q-function's overestimation bias [30, 31, 32, 33, 34], for better exploration [35, 36, 37], or for reducing bootstrap error propagation [38]. The closest to our approach is Anschel et al. [32] that stabilizes Q-learning by using the average of previously learned Q-values as the target Q-value. While prior works mostly focus on online RL and estimate the ground-truth Q-functions, we leverage an ensemble of pessimistically pre-trained Q-functions for safe offline-to-online RL.

# 6 Experiments

We designed our experiments to answer the following questions:

- How does our method compare to existing offline-to-online RL methods and an online RL method that learns from scratch (see Figure 3)?
- Can our balanced replay scheme locate offline samples relevant to the current policy (see Figure 4a) and improve the fine-tuning performance by utilizing these samples (see Figure 4c)?
- Can our pessimistic Q-ensemble scheme discriminate unseen actions (see Figure 4b) and successfully stabilize the fine-tuning procedure by mitigating distribution shift (see Figure 4d)?
- Does our method scale to vision-based robotic manipulation tasks (see Figure 5)?

## 6.1 Locomotion Tasks

**Setup.** We consider MuJoCo [11] locomotion tasks, i.e., `halfcheetah`, `hopper`, and `walker2d`, from the D4RL benchmark suite [12]. To demonstrate the applicability of our method on various suboptimal datasets, we use four dataset types: `random`, `medium`, `medium-replay`, and `medium-expert`. Specifically, `random` and `medium` datasets contain samples collected by a random policy and a medium-level policy, respectively. `medium-replay` datasets contain all samples encountered while training a medium-level agent from scratch, and `medium-expert` datasets contain samples collected by both medium-level and expert-level policies. For our method, we use ensemble size $N = 5$. More experimental details are provided in Section D.

**Comparative Evaluation.** We consider the methods outlined below as baselines for comparative evaluation. For fair comparison, we applied ensemble to all baselines except SAC-ft, since the results for SAC-ft with ensemble can be found in the ablation studies (see Figure 4c).

- Advantage Weighted Actor Critic [AWAC; 25]: an offline-to-online RL method that trains the policy to imitate actions with high advantage estimates.

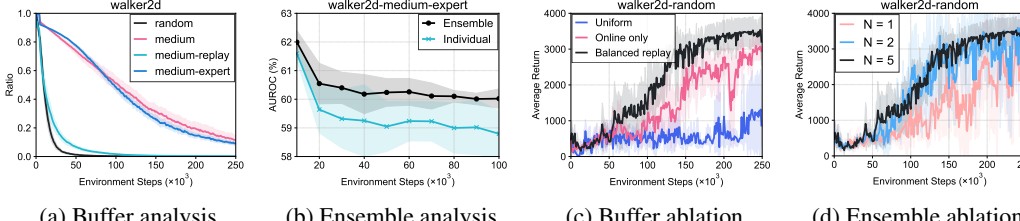

|(a) Buffer analysis|(b) Ensemble analysis|(c) Buffer ablation|(d) Ensemble ablation|

Figure 4: (a) Proportion of offline samples used for updates as the agent is fine-tuned online, for `walker2d` tasks. (b) AUROC (%) over the course of fine-tuning on `walker2d-medium-expert`, where the Q-function is interpreted as a binary classifier that classifies a given state-action pair $(s, a)$ as either a seen pair $(s, a_{\mathtt{seen}})$ or an unseen pair $(s, a_{\mathtt{uniform}})$, for a state $s$ encountered online. Pessimistic Q-ensemble shows a stronger discriminative ability. (c) Performance on `walker2d-random` with and without balanced experience replay. We consider two setups where balanced experience replay is not used: (i) Uniform, where offline and online samples are sampled uniformly from the same buffer for updates, and (ii) Online only, where the offline agent is fine-tuned using online samples only. (d) Performance on `walker2d-random` with varying ensemble size $N \in \{1, 2, 5\}$. The solid lines and shaded regions represent mean and standard deviation, respectively, across four runs.

- BCQ-ft: Batch-Constrained deep Q-learning [BCQ; 2], is an offline RL method that updates policy by modeling the behavior policy using a conditional VAE [39]. We extend BCQ to the online fine-tuning setup by applying the same update rules as offline training.
- SAC-ft: Starting from a CQL agent trained via (3, 2), we fine-tune the agent via SAC updates (1, 2). Justification for excluding the CQL regularization term from (3) during fine-tuning can be found in Section E.1.
- SAC: a SAC agent trained from scratch via (1, 2), i.e., the agent has no access to the offline dataset. This baseline highlights the benefit of offline-to-online RL, as opposed to fully online RL, in terms of sample efficiency.

Figure 3 shows the performances of our method and baseline methods considered during the online RL phase. In most tasks, our method outperforms the baseline methods in terms of both sample-efficiency and final performance. In particular, our method significantly outperforms SAC-ft, which shows that balanced replay and pessimistic Q-ensemble are indeed essential.

We also emphasize that our method performs consistently well across all tasks, while the performances of AWAC and BCQ-ft are highly dependent on the quality of the offline dataset. For example, we observe that AWAC and BCQ-ft show competitive performances in tasks where the datasets are generated by high-quality policies, i.e., `medium-expert` tasks, but perform worse than SAC on `random` tasks. This is because AWAC and BCQ-ft employ the same regularized, pessimistic update rule for offline and online setups alike, either explicitly (BCQ-ft) or implicitly (AWAC), which leads to slow fine-tuning. Our method instead relies on pessimistic *initialization*, and hence enjoys much faster fine-tuning, while not sacrificing the initial training stability.

**Balanced replay analysis.** To investigate the effectiveness of our balanced experience replay scheme for locating near-on-policy samples in the offline dataset, we report the ratios of offline samples used for updates fine-tuning proceeds. Figure 4a shows that for the `random` task, offline samples quickly become obsolete, as they quickly become irrelevant to the policy being fine-tuned. However, for the `medium-expert` task, offline samples include useful expert-level transitions that are relevant to the current policy, hence are replayed throughout the online training. This shows that our balanced replay scheme is capable of utilizing offline samples only when appropriate.

**Q-ensemble analysis.** We quantitatively demonstrate that pessimistic Q-ensemble indeed provides more discriminative value estimates, i.e., having distinguishably lower Q-values for unseen actions than for seen actions. In particular, we consider a `medium-expert` dataset, where the offline data distribution is narrow, and the near-optimal offline policy can be brittle. Let $\mathcal{D}_T^{\mathtt{real}} := \{(s_i, a_i)\}_{i=1}^T$ be the samples collected online up until timestep $T$. We construct a "fake" dataset by replacing the actions in $\mathcal{D}_T^{\mathtt{real}}$ with random actions, i.e., $\mathcal{D}_T^{\mathtt{fake}} := \{(s_i, a_{\mathtt{unif}})\}_{i=1}^T$. Interpreting $Q(s, a)$ as the confidence value for classifying real and fake transitions, we measure the area under ROC (AUROC) curve values over the course of fine-tuning. As seen in Figure 4b, Q-ensemble demonstrates superior discriminative ability, which leads to stable fine-tuning.

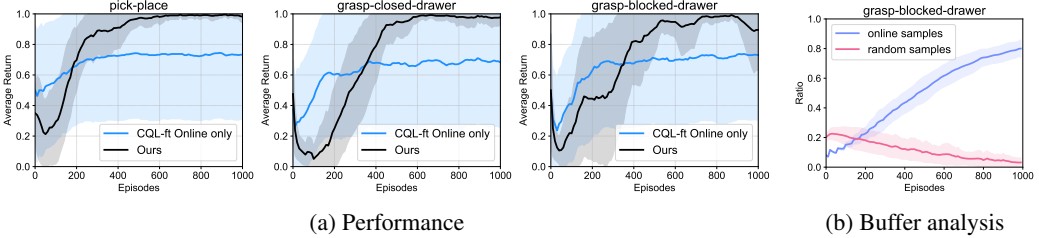

|  (a) Performance | (b) Buffer analysis |

Figure 5: (a) Fine-tuning performance for robotic manipulation tasks considered. (b) Proportion of random data used for during fine-tuning decreases over time. The solid lines and shaded regions represent mean and standard deviation, respectively, across eight runs.

**Ablation studies.** Figure 4c shows that balanced replay improves fine-tuning performance by sampling near-on-policy transitions. On the other hand, two other naïve sampling schemes – (a) Uniform, where offline and online samples are sampled uniformly from the same buffer, and (b) Online only, where the offline agent is fined-tuned using online samples exclusively – suffer from slow and unstable improvement, even with pessimistic Q-ensemble. This shows that balanced replay is crucial for reducing the harmful effects of distribution shift.

Also, Figure 4d shows that fine-tuning performance improves as the ensemble size $N$ increases, which shows that larger ensemble size provides higher-resolution pessimism, leading to more stable policy updates. Ablation studies for all tasks can be found in Section E.1.

### 6.2 Robotic Manipulation Tasks

**Setup.** We consider three sparse-reward pixel-based manipulation tasks from Singh et al. [10]: (1) `pick-place`: pick an object and put it in the tray; (2) `grasp-closed-drawer`: grasp an object in the initially closed bottom drawer; (3) `grasp-blocked-drawer`: grasp an object in the initially closed bottom drawer, where the initially open top drawer blocks the handle for the bottom drawer. Episode lengths for the tasks are 40, 50, 80, respectively.

The original dataset [10] for each task consists of scripted exploratory policy rollouts. For example, for `pick-place`, the dataset contains scripted pick attempts and place attempts. However, it is rarely the case that logged data 'in the wild' contains such structured, high-quality transitions only. We consider a more realistic setup where the dataset also includes undirected, exploratory samples – we replace a subset of the original dataset with uniform random policy rollouts. Note that random policy rollouts are common in robotic tasks [40, 41]. We used ensemble size $N = 4$ for our method. More details about the tasks and dataset construction are provided in Section D.2.

**Comparative Evaluation.** We compare our method with the method considered in Singh et al. [10], namely, CQL fine-tuning with online samples only. CQL-ft fails to solve the task in some of the seeds, resulting in high variance as seen in Figure 5a. This is because CQL shows inconsistent offline performance across random seeds due to such factors as difficulty of training on mixture data [12], instability of offline agents over stopping point of training [42], and sparsity of rewards. With no access to (pseudo-)expert offline data and due to heavy regularization of CQL, such CQL agents hardly improve. Meanwhile, our method consistently learns to perform the task within a reasonable amount of additional environment interaction (40K to 80K steps).

**Buffer analysis.** We analyze whether balanced replay scales to image-based robotic tasks. As seen in Figure 5b, without any privileged information, balanced replay automatically selects relevant offline samples for updates, while filtering out task-irrelevant, random data as fine-tuning proceeds.

## 7 Conclusion

In this paper, we identify state-action distribution shift as the major obstacle in offline-to-online RL. To address this, we present a simple framework that incorporates (1) a balanced experience replay scheme, and (2) a pessimistic Q-ensemble scheme. Our experiments show that the proposed method performs well across many continuous control robotic tasks, including locomotion and manipulation tasks. We expect our method to enable more sample-efficient training of robotic agents by leveraging offline samples both for offline and online learning. We also believe our method could prove to be useful for other relevant topics such as scalable RL [43] and RL safety [44].

**Acknowledgments**

This work was supported by Microsoft and Institute of Information & communications Technology Planning & Evaluation (IITP) grant funded by the Korea government(MSIT) (No.2019-0-00075, Artificial Intelligence Graduate School Program(KAIST)). We would like to thank anonymous reviewers for providing helpful feedbacks and suggestions in improving our paper.

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
