# OpenReview forum: "Offline-to-Online Reinforcement Learning via Balanced Replay and Pessimistic Q-Ensemble"
_robot-learning.org/CoRL/2021/Conference — CoRL2021 Poster_

### Official Review · Reviewer_yjtU · 2021-07-21

**Originality:** Good
**Technical Quality:** Excellent
**Clarity Of Presentation:** Excellent
**Impact:** 3

**Recommendation:**

Weak Accept: I recommend accepting the paper, but will not argue for my recommendation if the majority of other reviewers have a different opinion.

**Summary:**

The paper presents a method for off-line to on-line reinforcement learning. First, given a dataset of behavior in a task, an ensemble of "pessimistic" Q-functions and policies is trained off-line where the Q-function is learned conservatively to avoid over-estimation of Q-values in out of distribution state-action samples. Subsequently, on-line fine-tuning is done using a Balanced Experience Replay. This replay buffer consists of old samples from the off-line learning phase and new samples collected during on-line learning. Transitions are sampled based on their "online-ness", a value proportional to the density ratio between on-line samples and off-line samples. The further traning progresses and moves away from the initial data, the fewer off-line samples are chosen to update the policy.
The method is evaluated on simulated locomotion tasks from open-ai gym, as well as vision based manipulation tasks. It performs especially well in terms of learning speed if the intial policy is random.

**Issues:**

- I don't fully get the definition of the behavior policy in line 69. It looks like it is the number of times an action a_0 is taken in state s_0 divided by how often state s_0 occurs in the replay buffer. But, given states and actions are continuous, how often will this be different from 1?
- Why is the performance of half cheetah worse, if the policy is initially trained with the medium dataset?
- Why does the performance using the medium-expert dataset on hopper and walker2d not reach the performance of AWAC and partially BCQ-ft?
- Is there a difference in the entropy of the policy after off-line training with a random, medium or expert data set? For example does the policy not explore enough if trained on an expert data set?
- Why are the manipulation tasks only compared to [10]?



**Reviewer Expertise:**

Fair: Some knowledge of the area

**Strengths And Weaknesses:**

Strenghts:
- The overall method seems very intuitive and well motivated
- The efficacy of each component is shown via ablation studies

Weaknesses:
- Especially on halfcheetah, the asymptotic performance seems to stay well below what SAC achieves alone (12k vs 15k). An intuition why this is the case would be interesting.

**Summary Of Recommendation:**

First of all, I am not an expert in off-line RL, so I'm not familiar with all the work that is out there on the topic. Most of the parts like CQL, ensembles, and prioritzed replay based on likelihoods is not entirely new but the parts are modified nicely to fit the off-line to on-line RL scheme and put together well and results look promising. Therefore, I would recommend accept but put my confidence score rather low.

---

> ### Author Response · Authors · 2021-08-23
> **Response to Reviewer yjtU (1/2)**
>
> Dear Reviewer yjtU,
>
> We sincerely appreciate your valuable comments and efforts helpful for improving our manuscript. We address each comment in detail, one by one as below.
>
> ---
>
> **Q1. Especially on halfcheetah, the asymptotic performance seems to stay well below what SAC achieves alone (12k vs 15k). An intuition why this is the case would be interesting.**
>
> **A1.** We used [rlkit](https://github.com/rail-berkeley/rlkit) for our implementation where SAC reaches the asymptotic performance of 12k ~ 13k for the HalfCheetah-v2 environment (see a relevant [discussion](https://github.com/rail-berkeley/rlkit/issues/128)). Also, note that D4RL [1] halfcheetah expert trajectories also achieve the score of 12k.
> Hence, we emphasize that our results on halfcheetah-random and halfcheetah-medium-expert achieves similar asymptotic performance to SAC, while interacting with the environment for 0.25 mil steps -- only a fraction of samples required to reach asymptotic performance for halfcheetah, 3 mil. As for the performance on halfcheetah-medium, see our response to **Q2**.
>
> ---
>
>
> **Q2. Why is the performance of halfcheetah worse, if the policy is initially trained with the medium dataset?**
>
> **A2.** Thank you for pointing this out. We also noticed this while preparing for the initial draft, and in order to investigate it further, we additionally ran our method on halfcheetah tasks for 1mil steps. As shown in Figure 13 of the revised manuscript, our method performed best on random dataset, reaching 15k average return, while reaching only \~12k for medium and medium-replay datasets. We conjecture this is because the agent sees the most diverse set of samples when trained from scratch, and thus its Q-function extrapolates better, leading to more efficient exploration. While the medium-replay dataset also contains diverse samples, it asymptotically performs worst possibly because it contains significantly less data compared to the other datasets -- random and medium contain 1mil transitions, respectively, and medium-expert contains 2mil transitions. We provided a discussion on this in Section E.1 of the revised manuscript.
>
> ---
>
> **Q3. Is there a difference in the entropy of the policy after offline training with a random, medium or expert dataset?**
>
> **A3.** We provide a plot for expected policy entropy over the visited states during fine-tuning in the following [link](https://imgur.com/a/LcToSjh).
> Indeed, the starting policy entropies are different; in particular, highest for halfcheetah-random, naturally so given the high-entropy behavior policy used to generate the data. However, they quickly converge to the same target entropy value of -6.0, due to use of automatic temperature tuning in SAC [2], and it is difficult to draw a meaningful observation from this.
> Having said that, we do agree that dataset composition may have a profound impact on the agent’s ability to explore, and a relevant discussion was provided in response to **Q2**.
>
> ---
>
> **Q4. I don’t fully get the definition of the behavior policy in line 69.**
>
> **A4.** The definition is from the original CQL paper [3]. As pointed out, when the state and action spaces are continuous, there is only one such action for a given state in the dataset almost surely. In practice, this just means that when one calculates the term
>
> $ \mathbb{E}_{s \sim \mathcal{B}, a \sim {\hat{\pi}}_\beta }[Q(s,a)], $
>
> one samples a state-action pair from the offline dataset.
>
> ---
>
> **Q5. Why does the performance using the medium-expert dataset on hopper and walker2d not reach the performance of AWAC and partially BCQ-ft?**
>
>
> **A5.** AWAC and BCQ essentially perform (selective) imitation learning via constrained updates. Therefore when the dataset contains a significant amount of expert trajectories, they tend to perform well. However, note that our method still performs almost on par with AWAC and BCQ-ft in medium-expert, and outperforms them significantly in most other setups.
>
> ---
>
> **Q6. Why are the manipulation tasks only compared to [4]?**
>
> **A6.** For the D4RL MuJoCo dataset [1], baseline methods such as AWAC and BCQ are well-tested, at least in the offline setup. However, for the manipulation tasks from COG [4], no methods have been tested other than COG and naive baselines such as SAC followed by behavior cloning initialization. As a preliminary experiment, we performed experiments with BC + SAC, which is shown to significantly underperform our method even for the easiest task of pick-place (see [here](https://imgur.com/a/gzi2sBG) for the results). We will continue our effort to provide results for other baselines during the discussion phase or in the final draft.

---

> > ### Author Response · Authors · 2021-08-23
> > **Response to Reviewer yjtU (2/2)**
> >
> > ---
> > **References**
> >
> > [1] J. Fu, A. Kumar, O. Nachum, G. Tucker, and S. Levine.  D4rl:  Datasets for deep data-driven reinforcement learning. arXiv preprint arXiv:2004.07219, 2020.
> >
> > [2] Haarnoja, T., Zhou, A., Hartikainen, K., Tucker, G., Ha, S., Tan, J., Kumar, V., Zhu, H., Gupta, A., Abbeel, P. and Levine, S., 2018. Soft actor-critic algorithms and applications. arXiv preprint arXiv:1812.05905.
> >
> > [3] A. Kumar, A. Zhou, G. Tucker, and S. Levine.  Conservative q-learning for offline reinforce-ment learning. NeurIPS, 2020
> >
> > [4] A. Singh, A. Yu, J. Yang, J. Zhang, A. Kumar, and S. Levine.  Cog: Connecting new skills to past experience with offline reinforcement learning. CoRL, 2020.

---

> > ### Comment · Reviewer_yjtU · 2021-09-06
> > **Response to Author Rebuttal**
> >
> > This comment is to acknowledge that I have read the authors' revisions and responses to the reviewers. The authors did a good job in answering my questions. After reading the other reviewers comments, my recommendation for weak accept stays.

---

### Official Review · Reviewer_Jt9W · 2021-07-21

**Originality:** Very Good
**Technical Quality:** Very Good
**Clarity Of Presentation:** Good
**Impact:** 4

**Recommendation:**

Weak Accept: I recommend accepting the paper, but will not argue for my recommendation if the majority of other reviewers have a different opinion.

**Summary:**

This paper identifies state-action distribution shift as the major problem in offline-to-online RL, and proposes two mechanisms, balanced replay buffer and pessimistic Q-ensemble to solve it. Through well-organized experiments, the paper shows that the balance experience replay can effectively balance the usage of online data and offline data and the pessimistic Q-ensemble is able to mitigate distributions shift.  The proposed method obtains good empirical results on D4RL Mujoco locomotion tasks and vision-based robotic manipulation tasks.

**Issues:**

1. In section 3.3, the author compares the choice of CQL and FQE and shows that the CQL is more suitable for offline-to-online RL. But according to the CQL paper, for many offline datasets, directly using non-pessimistic Q-function suffers from distribution shift and can’t learn well during offline training. Thus if possible, I think it will be good to compare CQL with another offline RL method, e.g. BCQ etc.

2. In section 4.1, the training details are not very clear for me. For example, the author uses P, Q to represent d^{off} and d^{on}, thus the density ratio w(s,a):=d^{on}(s,a)/d^{off}(s,a) = Q/P, but in the last sentence of page 4, the author uses ”estimate the density ratio dP/dQ with a parametric model w_{phi}(x)”, which is inconsistent.

**Reviewer Expertise:**

Good: General knowledge of the area

**Strengths And Weaknesses:**

Strengths:
1. The motivation of this paper is strong and the proposed method effectively solves the identified state-action distribution shift problem.
2. The empirical results are good. It outperforms previous methods in most cases.
3. The paper is well written and easy to follow.
4. The experiments are well-designed and show the benefits of the proposed methods.

Weaknesses:
1. In section 3.3, the author compares the choice of CQL and FQE and shows that the CQL is more suitable for offline-to-online RL. But according to the CQL paper, for many offline datasets, directly using non-pessimistic Q-function suffers from distribution shift and can’t learn well during offline training. Thus, I think it will be good to compare CQL with another offline RL method, e.g. BCQ etc.
2. In section 4.1, the training details are not very clear for me. For example, the author uses P, Q to represent d^{off} and d^{on}, thus the density ratio w(s,a):=d^{on}(s,a)/d^{off}(s,a) = Q/P, but in the last sentence of page 4, the author uses ”estimate the density ratio dP/dQ with a parametric model w_{phi}(x)”, which is inconsistent.


**Summary Of Recommendation:**

My recommendation is "Weak Accept".

This paper identifies state-action distribution shift as the major problem in offline-to-online RL, and proposes two mechanisms, balanced replay buffer and pessimistic Q-ensemble to solve it. The motivation of this paper is strong and the proposed method effectively solves the identified state-action distribution shift problem. The proposed method obtains good empirical results on D4RL Mujoco locomotion tasks and vision-based robotic manipulation tasks. But the description of the training details can be more clear. Thus, my recommendation is "Weak Accept".

---

> ### Author Response · Authors · 2021-08-23
> **Response to Reviewer Jt9W**
>
> Dear Reviewer Jt9W,
>
> We sincerely appreciate your helpful comments for improving our manuscript. We address each comment in detail, one by one as below.
>
> ---
>
> **Q1. In section 3.3, the authors compare the choice of CQL and FQE and show that the CQL is more suitable for offline-to-online RL. But according to the CQL paper, for many offline datasets, directly using non-pessimistic Q-function suffers from distribution shift, and can’t learn well during offline training.**
>
> **A1.** We would like to clarify that the offline training for both FQE-ft and CQL-ft are done via CQL (as indicated in lines 118-122 of the manuscript), but they differ in that CQL-ft fine-tunes the Q-function obtained via offline training, whereas FQE-ft fine-tunes a newly trained, neutral Q-value function for the policy obtained via FQE. What we showed is that even with successful offline training (hence a good offline policy), a non-pessimistic Q-function can lead to distribution shift during fine-tuning. We further clarified this detail.
>
> ---
>
> **Q2. In section 4.1, the training details are not very clear for me. For example, the author uses P, Q to represent $d^{\texttt{off}}$ and $d^{\texttt{on}}$, thus the density ratio $w(s,a):=d^{\texttt{on}}(s,a)/d^{\texttt{off}}(s,a) = Q/P,$ but in the last sentence of page 4, the author uses ”estimate the density ratio $dP/dQ$ with a parametric model $w_{\phi}(x)$”, which is inconsistent.**
>
> **A2.** We are very grateful for this catch. $P$ and $Q$ should be probability distributions whose densities are given by $d^{\texttt{on}}$, $d^{\texttt{off}}$, respectively. We corrected the mistakes in notations and revised the manuscript accordingly. Please also refer to [1] and [2] for a more detailed explanation.
>
> ---
>
> **References**
>
> [1] X. Nguyen, M. J. Wainwright, and M. I. Jordan.  Estimating divergence functionals and the likelihood ratio by penalized convex risk minimization. NeurIPS, 2008.
>
> [2] S. Sinha, J. Song, A. Garg, and S. Ermon.  Experience replay with likelihood-free importance weights, 2021. URL https://openreview.net/forum?id=ioXEbG_Sf-a.

---

> > ### Comment · Reviewer_Jt9W · 2021-09-03
> > **Reviewer comment**
> >
> > Thanks for the author's detailed answer. My questions are answered, but after reading the author's responses to other reviewers, I still have one main concern about this paper. The paper argued that the performance drop in medium-expert datasets and manipulation tasks is because of the narrow data distribution (see the response to reviewer rPXZ Q1). This is one possible reason, but it might be also caused by the change of the objective function of the Q. Overall, this paper points out the distribution shift problem in offline-to-online RL and the proposed method shows its advantages via lots of experiments. But consider that the existence of the instability problem, I will keep my recommendation as **Weak Accept**.

---

### Official Review · Reviewer_rPXZ · 2021-07-23

**Originality:** Good
**Technical Quality:** Good
**Clarity Of Presentation:** Good
**Impact:** 3

**Recommendation:**

Weak Accept: I recommend accepting the paper, but will not argue for my recommendation if the majority of other reviewers have a different opinion.

**Summary:**

This main contribution of this paper is a balanced replay scheme to balance the offline and online data. It uses an offline policy evaluation method to compute the importance sampling ratio between offline samples and online samples, which is further used to prioritize experience replay of offline samples. The motivation is clear and experimental results are promising. Overall, it's a good paper.

**Issues:**

As mentioned in "Strengths and Weaknesses".

**Reviewer Expertise:**

Excellent: Expert knowledge on the topic of the paper

**Strengths And Weaknesses:**

Strengths:
  (1) The idea of Leveraging offline samples which relate to the current policy to boost online policy evaluation and improvement is reasonable since offline dataset may consist of samples with different styles and quality, some of which may harm the currently learned policy, let alone the policy improvement.
  (2) Extensive experimental results demonstrate the effectiveness of the proposed balanced replay scheme.


Weaknesses:
(1) Though the authors claim that their method can prevent the policy from being destroyed by online OOD samples, we can still observe a severe performance drop at the early period of online interaction in most cases, why?
(2) While pessimism protects the policy from severe bootstrapping error,  it may also prevent the online policy from efficiently exploring the unknown environment (in fact, there have been many works using pessimism for online exploration). In other words, the online policy may be too conservative for online policy improvement. I think it's necessary to provide such ablation studies telling if pessimism is helpful for the ultimate online policy performance

**Summary Of Recommendation:**

Overall, this is a good paper. But If the authors could provide with more experiments to clarify the two weaknesses, I will raise my score.

---

> ### Author Response · Authors · 2021-08-23
> **Response to Reviewer rPXZ**
>
> Dear Reviewer rPXZ,
>
> We sincerely appreciate your valuable comments for improving our manuscript. We address each comment in detail, one by one as below.
>
> ---
>
> **Q1. Though the authors claim that their method can prevent the policy from being destroyed by online OOD samples, we can still observe a severe performance drop at the early period of online interaction in most cases. Why?**
>
> **A1.** As the offline data distribution becomes narrow, it becomes much easier for the agent to drift away from the seen, familiar state-action regime. For this reason, we observe performance drop due to distribution shift for e.g., (1) medium-expert MuJoCo tasks, where highly selective behavior policy leads to a narrow coverage of the state-action space; and (2) manipulation tasks, where high-dimensional, pixel observation space implies only a narrow coverage of the state-action space by the offline dataset. However, note that only our method succeeds to recover the good performance quickly in most cases where there is a performance degradation (with the exception of hopper-medium-expert, 10K~20K steps suffice for recovery for both MuJoCo and manipulation domains). This problem could potentially be solved by regularizing the policy to be close to the behavior policy (e.g., a behavior cloning term), and would be an interesting future direction to explore.
>
> ---
>
> **Q2. While pessimism protects the policy from severe bootstrapping error, it may also prevent the online policy from efficiently exploring the unknown environment. In other words, the online policy may be too conservative for online policy improvement. I think it’s necessary to provide such ablation studies telling if pessimism is helpful for the ultimate online policy performance.**
>
> **A2.** Thank you for pointing this out. We performed additional experiments in the halfcheetah domain -- where the trends are most visible -- in order to compare the asymptotic performances of FQE-ft (neutral Q-function) and CQL-ft (pessimistic Q-function). As seen in Figure 13 of the revised manuscript, we observed that CQL-ft and FQE-ft had more or less similar asymptotic performances with the exception of halfcheetah-random, where CQL-ft noticeably outperforms FQE-ft. This shows that initial pessimism does not harm the asymptotic online performance, for our method gradually loses such pessimism over the course of fine-tuning. A relevant discussion can be found in Section E.1 of the revised manuscript.

---

### Official Review · Reviewer_SrrQ · 2021-07-25

**Originality:** Good
**Technical Quality:** Good
**Clarity Of Presentation:** Very Good
**Impact:** 3

**Recommendation:**

Weak Accept: I recommend accepting the paper, but will not argue for my recommendation if the majority of other reviewers have a different opinion.

**Summary:**

The authors attempted to address the challenge posed by distribution shift to fine tuning policies learned through offline RL. The proposed solution includes two parts: (1) a method called “balanced replay” that uses a density approximation of how similar the old offline samples are to the new online samples and accordingly determines how offline and online samples in the replay buffer should be mixed during fine tuning; and (2) an ensemble of “pessimistic” Q functions (trained with conservative deep Q learning) to further strengthen the stability of the fine tuning.

**Issues:**

Technical issues

i.	Section 3 and Figure 1: Can the discussion and presentation here be used to motivate the necessity of the use of ensemble in the solution resolution? How should the discussion and Figure 1 (or experiments therein) be changed?

ii.	Should we also consider offline RL approaches (e.g. model-based) that learn the structure more explicitly before fine-tuning? How would that affect the distribution shift situation? Should we also consider model-based baselines? (No need to actually include such baselines. Just theoretically speaking.)

iii.	How does our analysis (e.g. in Section 3) inform about data collection & selection for the offline phase itself? Should we also compare with approaches that do data collection differently for the offline phase to improve fine-tuning? Note that the injection of uniform random policy rollouts (Line 291) seems to be an example. Should we also consider this in our theoretical treatment at least?

iv.	Line 155: In the design space here, other than d^on / d^off used in the paper, d^on / d^(on union off) also seems a sensible choice that might have the added benefit of trending towards 1 asymptotically.

v.	“Ensemble methods” (paragraph starting on Line 207): it is not clear that “obtain a high-resolution pessimism” is mutually exclusive. I.e. it is not clear “instead” is warranted.

vi.	Line 236-237: Is it justified to apply the same update rules as offline training when using BCQ as a baseline? Why not transfer/fine-tune using e.g. SAC? That seems to be a much more fair comparison. This is so especially in light of Figure 3 in which BCQ is barely learning during online fine-tuning. But BCQ was not designed for fine tuning. To do justice to BCQ, the comparison should presumably be about the quality of policy initialization.

vii.	Line 268: “Interpreting Q(s, a) as the confidence value for classifying real and fake transitions” -- How is this justifiable at all? Are we making a category mistake here? Action value and confidence about real vs. fake are radically different kinds of things.

viii.	Figure 5: “eight runs” and everywhere else is 4 runs. Why do we need to go up to eight runs here? What’s the reason behind this shift from the practice elsewhere in the paper.

ix.	Figure 13 and 14: the main expected effect or phenomenon -- stable and efficient fine-tuning -- seems highly unstable. All methods see big drops during the first 100 episodes. This raises questions about whether the method is going to be truly effective when used with physical robots. Of course, an alternative conclusion here could be that the method is good but that these environments are inadequate for testing the new architecture. It seems that adequate discussion of these curves is missing.

x.	Line 291: “we replace a subset of the original dataset with uniform random policy rollouts.” Is this justified? In real world applications, data are more likely from medium-level policies, less likely from uniform random policies. In fact, it could be the case that in the real world it’s much harder to obtain data with uniform random policy rollouts. That this sort of uniform random rollouts are needed at all seems worrying, especially if offline RL is supposed to use real world data.

Writing issues

i.	Line 83-84 “dataset of random halfcheetah transitions”. The notion of a “random dataset”  should be more appropriately introduced. More detail was provided later around Line 123. But it is more desirable if the notion is explained before it is used.

ii.	Line 131-133: It may be more helpful to say right there and then positively and explicitly that that’s what the balanced replay approach is intended to do. That wrap up Section 3 more adequately.

iii.	Figure 2: Diagram is good. But it is misleading to have labels of the parts of the diagram in shaded boxes. Consider remove the boxes around “Pessimistic Online Training”, “Environment Interaction” and “Balanced Replay”.

iv.	Line 196: the acronym AWAC should be introduced here rather than later on Line 233.

v.	Figure 1 and Figure 5: It will be helpful and make the figures more self-contained if the environments used in the experiments are explicitly identified in the caption.

vi.	Figure 13: should “CQL objective” in the caption be “SAC objective”?


**Reviewer Expertise:**

Good: General knowledge of the area

**Strengths And Weaknesses:**

Strengths

i.	The case for the need to address distribution shift in fine tuning policies learned through offline RL is well motivated: Section 3 and Figure 1 are well structured and well presented. Kudos to the authors!

ii.	Balanced replay is a relatively novel approach.

iii.	The ensemble of pessimistic Q functions is empirically effective.

Weaknesses

i.	The relationship between the two parts of the solution is unclear. The pessimistic ensemble seems to be an extra add-on. It seems to be an afterthought for better performance  rather than an integrated part of an architecturally cohesive solution.

ii.	Discussion of the ensemble, about why and how it works, stopped at vague allusions to “resolution” without detail. (Technical Issue v below).

iii.	The treatment of BCQ as a baseline seems quite unfair. See Technical Issues (vi) below for more detail.

iv.	The effectiveness of the method in the real world is doubtful. See Technical Issues (ix and x) below for more detail.


**Summary Of Recommendation:**

a.	There is an interesting idea that is worth discussing by the RL community: “balanced replay”. Also, the analysis of distribution shift through a careful study of one concrete case is quite helpful and illuminating.

b.	The relationship between the pessimistic ensemble and distribution shift is not clearly established. From the experiments, it seems that the ensemble is mainly contributing towards stabilizing training. Overall, the ensemble part seems to be a separate proposal.

c.	The effectiveness of the approach for real-world robotics is doubtful. See below in "Issues".

d.	There are multiple technical issues that need to be fixed, addressed or discussed. See below in “Issues”.

---

> ### Author Response · Authors · 2021-08-23
> **Response to Reviewer SrrQ (1/3)**
>
> Dear Reviewer SrrQ,
>
> We sincerely appreciate your valuable and insightful comments. We found them very helpful for improving our manuscript. We address each comment in detail, one by one below.
>
> ---
>
> **Q1. Section 3 and Figure 1: Can the discussion and presentation here be used to motivate the necessity of the use of ensemble in the solution resolution? How should the discussion and Figure 1 (or experiments therein) be changed?**
>
> **A1.** We argue that the discussion in section 3.3 naturally motivates our ensemble method, for it shows that leveraging a pessimistic Q-function is essential for preventing distribution shift, and ensembling such Q-functions provides a more accurate, “high-resolution” pessimism (please see our response to **Q2** for a clarification on what we mean by high-resolution pessimism). We made this more explicit in the revised manuscript.
>
> ---
>
> **Q2. “Interpreting Q(s,a) as the confidence value for classifying real and fake transitions” -- how is this justifiable at all? Are we making a category mistake here? Action value and confidence about real vs. fake are radically different kinds of things.**
>
> **A2.** As explained in Section 3, to prevent distribution shift, Q-values must be distinguishably lower for unseen actions than for seen actions during fine-tuning.
> While this is not strictly a classification problem, viewing Q-values for seen vs unseen actions as proxy for the agent’s high vs low confidence is a good way to quantitatively measure the discriminative ability of Q-function. Based on this, AUROC values during fine-tuning were provided thereof, and indeed, pessimistic Q-ensemble was shown to provide a more discriminative value estimate. By high-resolution pessimism, we were referring to such superior discriminative ability of the Q-ensemble.
>
> ---
>
> **Q3. “Ensemble methods” (paragraph starting on Line 207): it is not clear that “obtain a high-resolution pessimism” is mutually exclusive. I.e., it is not clear “instead” is warranted.**
>
> **A3.** Thank you for pointing this out. We initially wanted to emphasize that prior methods leverage Q-functions trained to estimate the ground-truth Q-functions; whereas our method leverages Q-functions pre-trained to estimate a lower bound of the ground-truth Q-function. However, we realize that various overestimation bias reduction methods [1-5] and SUNRISE [6] can be seen as performing a more conservative bellman backup. We reflected this point in the revised manuscript accordingly.
>
> ---
>
> **Q4. Why does BCQ use the same update rule as offline training and not transfer/fine-tune using SAC? Is this fair?**
>
> **A4.** We remark that the current comparison is fair due to the following reasons: BCQ crucially relies on a conditional VAE for action selection, and $\pi$ merely adds noise to the actions output by the VAE decoder. Hence, it is unnatural to apply SAC updates for a BCQ agent. Nair, et al. [7] also pointed out that such reliance on a behavior model (VAE) is a bottleneck in fine-tuning.
>
> Meanwhile, CQL objective is identical to the SAC objective except for the added CQL regularization term (and the maximum-entropy term for bellman backup) for policy evaluation, which makes it natural to exclude it during fine-tuning.

---

> > ### Author Response · Authors · 2021-08-23
> > **Response to Reviewer SrrQ (2/3)**
> >
> > ---
> > **Q5. Figure 13 and 14: the main expected effect or phenomenon -- stable and efficient fine-tuning -- seems highly unstable. All methods see big drops during the first 100 episodes. This raises questions about whether the method is going to be truly effective when used with physical robots. Of course, an alternative conclusion here could be that the method is good but that these environments are inadequate for testing the new architecture. It seems that adequate discussion of these curves is missing.**
> >
> > **A5.** The vision-based robotic manipulation tasks are challenging, for its high-dimensional, pixel observation space implies only a narrow coverage of the state-action space by the offline dataset. Additionally, these tasks only provide sparse rewards upon task completion. Hence, initial performance degradation is understandable during early exploration. However note that it only takes our method \~200 episodes (around 8~16K environment steps) to recover the initial score after an initial phase of exploration, and reaches high score in all cases asymptotically.
> >
> > Also, we note that for manipulation tasks, both ensemble and balanced replay were essential for good performance. First, due to reward sparsity, there were individual offline agents that, while producing meaningful trajectories that resemble some trajectories in the offline dataset, were nonetheless not good enough to receive reward signals online. In this case, the ensemble agent becomes more robust than these individual agents, and was more likely to see enough reward signals for fine-tuning. Second, even with ensemble, updating with online samples only results in severe distribution shift, and balanced replay was essential to facilitate fine-tuning.
> >
> > We added a relevant discussion in Section E.2 of the revised manuscript.
> >
> > ---
> >
> > **Q6. “We replace a subset of the original dataset with uniform random policy rollouts.” Is this justified? In real world applications, data are more likely from medium-level policies, less likely from uniform random policies. In fact, it could be the case that in the real world it’s much harder to obtain data with uniform random policy rollouts. That this sort of uniform random rollouts are needed at all seems worrying, especially if offline RL is supposed to use real world data.**
> >
> > **A6.** We would like to first point out that random policy rollouts were not *needed* to achieve good performance. Rather, we *replaced* half the original dataset with random policy rollouts, in order to showcase our method’s capability to retrieve relevant offline samples for fine-tuning.
> > Also, in many robotics tasks, samples generated by random policy provide various advantages: For instance, many prior works in robotic manipulation [8,9] collect random policy rollouts, for random policy rollouts (1) enable simple task/reward-agnostic exploration of the environment, and as a result (2) enable superior generalization and better performance in downstream tasks. All and all, it is a realistic scenario for offline datasets to contain random rollouts, and an offline-to-online RL method must be able to handle them just as well.
> >
> > ---
> >
> > **Q7. Line 155: In the design space here, other than $d^{\texttt{on}}/d^{\texttt{off}}$ used in the paper, $d^{\texttt{on}}/d^{\texttt{on}\cup\texttt{off}}$ also seems a sensible choice that might have the added benefit of trending towards 1 asymptotically.**
> >
> > **A7.** Thank you for the suggestion. We additionally experimented with this scheme, and provided the results in Section E.1 of the revised manuscript (Figure 14). With the same hyperparameters, we observed similar performance across most setups. However, as you mentioned, the alternative scheme may have the potential benefit of trending towards 1.0 asymptotically.
> >
> > ---
> >
> > **Q8 Should we also consider offline RL approaches (e.g., model-based) that learn the structure more explicitly before fine-tuning? How would that affect the distribution shift situation? Should we also consider model-based baselines? (No need to actually include such baselines. Just theoretically speaking)**
> >
> > **A8.** Dynamics model training does not involve regression to a bootstrapped target, hence may be easier to fine-tune -- possibly by oversampling near-online transitions via balanced replay, in order to enable timely fine-tuning. We expect that a model-based method such as MOPO [10], which also trains a pessimistic Q-function, may also be amenable to online fine-tuning. Looking into whether the fictitious samples generated near online, novel samples are reliable would be an interesting direction to explore.

---

> > > ### Author Response · Authors · 2021-08-23
> > > **Response to Reviewer SrrQ (3/3)**
> > >
> > > ---
> > > **Q9. How does our analysis (e.g. in Section 3) inform about data collection & selection for the offline phase itself? Should we also compare with approaches that do data collection differently for the offline phase to improve fine-tuning? Note that the injection of uniform random policy rollouts (Line 291) seems to be an example. Should we also consider this in our theoretical treatment at least?**
> > >
> > > **A9.** While our task of interest was offline-to-online RL with a dataset given a priori, it is certainly interesting to also consider the data collection process itself that would benefit downstream online tasks.
> > >
> > > Discussion from Section 3 suggests that diverse (and large) dataset with a wide coverage of the state-action space are less likely to lead to distribution shift.
> > >
> > > To further investigate the relationship between dataset composition and fine-tuning performance, we measured the online asymptotic performance of our method. As shown in Figure 13 of the revised manuscript, our method performed best for halfcheetah-random (15k score), while reaching suboptimal scores for medium and medium-replay tasks (12k score). We conjecture that observing a wide-support data may be essential not just for avoiding distribution shift, but also for asymptotic performance online.
> > >
> > > Given this, while the inclusion of uniform random policy rollouts in our robotic manipulation experiments was not for boosting performance (see response to **Q6**), it is likely that such random rollouts may benefit fine-tuning by providing a diverse set of exploratory transitions.
> > >
> > > ---
> > >
> > > **Q10. Figure 5: “Eight runs” and everywhere else is 4 runs.**
> > >
> > > **A10.** We wanted to show that the high variance of CQL-ft (Online only) does not come from the lack of random seeds. Trends are similar for 4 vs 8 random seeds, as seen in the following [plot](https://imgur.com/a/9YHjJWm).
> > >
> > > ---
> > >
> > > **Q11. Editorial comments**
> > >
> > > **A11.** We have addressed the issues and revised the manuscript accordingly (written in red in the revised manuscript). Thank you for your comments and suggestions.
> > >
> > > ---
> > >
> > > **References**
> > >
> > > [1] H. V. Hasselt. Double q-learning. NeurIPS, 2010.
> > >
> > > [2] H. v. Hasselt, A. Guez, and D. Silver. Deep reinforcement learning with double q-learning. AAAI, 2016.
> > >
> > > [3] O. Anschel, N. Baram, and N. Shimkin.  Averaged-dqn:  Variance reduction and stabilization for deep reinforcement learning. ICML, 2017.
> > >
> > > [4] S. Fujimoto, H. Hoof, and D. Meger.  Addressing function approximation error in actor-critic methods. ICML, 2018.
> > >
> > > [5]  Q. Lan, Y. Pan, A. Fyshe, and M. White.  Maxmin q-learning: Controlling the estimation bias of q-learning. ICML, 2020.
> > >
> > > [6] K.  Lee,  M.  Laskin,  A.  Srinivas,  and  P.  Abbeel.   Sunrise:  A  simple  unified  framework  for ensemble learning in deep reinforcement learning. ICML, 2021.
> > >
> > > [7] A. Nair, M. Dalal, A. Gupta, and S. Levine. Awac: Accelerating online reinforcement learning with offline datasets. arXiv preprint arXiv:2006.09359, 2021
> > >
> > > [8] C. Finn and S. Levine.  Deep visual foresight for planning robot motion. ICRA, 2017.
> > >
> > > [9] F. Ebert, C. Finn, S. Dasari, A. Xie, A. Lee, and S. Levine. Visual foresight: Model-based deep reinforcement  learning  for  vision-based  robotic  control. arXiv  preprint arXiv:1812.00568,4022018
> > >
> > > [10] T. Yu, G. Thomas, L. Yu, S. Ermon, J. Zou, S. Levine, C. Finn, and T. Ma. Mopo: Model-based offline policy optimization. NeurIPS, 2020.

---

> > > > ### Comment · Reviewer_SrrQ · 2021-09-04
> > > > **Response to Rebuttal**
> > > >
> > > > I would like to thank the authors for a thorough rebuttal. I found the explanation of the relationship between pessimistic Q ensemble and the balanced replay to be adequate. I also found the discussion about performance during initial online fine-tuning to be helpful. I have accordingly adjusted my recommendation.

---

### Author Response · Authors · 2021-08-23
**Summary of revisions**

Dear reviewers,

We sincerely appreciate your insightful comments and constructive suggestions to help us improve the manuscript. We are grateful for all positive comments: clear presentation (R1-4), well-motivated (R1-4), novelty (R1), and strong experimental results (R2-4).

In response to the questions and concerns raised, we have carefully revised and improved the manuscript with the following additional experiments and discussions:

- Improved figures for better readability (Figure 1, 2, 4)
- Clearer presentation on balanced replay (Section 4.1)
- Additional experiments and discussions regarding the relationship between dataset composition and performance (Section E.1, Figure 13)
- Additional experiments with an alternative density ratio (Section E.1, Figure 14)
- More detailed discussion on robotic manipulation task results (Section E.2)
- Other minor revisions (clearer motivation for method, clarification on prior works, typo corrections, etc.)

The revisions made are marked with “red” in the revised manuscript.

We also appreciate your continued effort to provide further feedback until the very end of response/discussion phase. We will make sure to reflect the comments in the final version.

Best regards,

Authors.

---

### Meta-Review · Area_Chair_yF7o · 2021-08-06

**Recommendation:** Accept (Poster)
**Confidence:** 4

**Metareview:**

The reviewers generally appreciate the strengths of the paper, with a well-motivated and mostly novel approach to offline RL. I am glad the authors were active during the rebuttal period and were able to address many reviewer concerns. I note that Reviewer Jt9W still has one concern about the method; namely, I advise the authors to add a paragraph or two conceding and elaborating on potential shortcomings of their proposed approach, including the potential instability issues.

---

### Decision · Program_Chairs · 2021-09-13

**Decision:**

Accept (Poster)

**Comment:**

The reviewers generally appreciate the strengths of the paper, with a well-motivated and mostly novel approach to offline RL. I am glad the authors were active during the rebuttal period and were able to address many reviewer concerns. I note that Reviewer Jt9W still has one concern about the method; namely, I advise the authors to add a paragraph or two conceding and elaborating on potential shortcomings of their proposed approach, including the potential instability issues.